# An Optical Device Based on a Chemical Chip and Surface Plasmon Platform for 2-Furaldehyde Detection in Insulating Oil

**DOI:** 10.3390/s24165261

**Published:** 2024-08-14

**Authors:** Letizia De Maria, Francesco Arcadio, Giuseppe Gabetta, Daniele Merli, Giancarla Alberti, Luigi Zeni, Nunzio Cennamo, Maria Pesavento

**Affiliations:** 1RSE S.p.A., Via Rubattino 54, 20134 Milan, Italy; letizia.demaria@rse.web.it; 2Department of Engineering, University of Campania L. Vanvitelli, Via Roma 29, 81031 Aversa, Italy; francesco.arcadio@unicampania.it (F.A.); luigi.zeni@unicampania.it (L.Z.); nunzio.cennamo@unicampania.it (N.C.); 3CESI S.p.A., Via Rubattino 54, 20134 Milan, Italy; giuseppe.gabetta@cesi.it; 4Department of Chemistry, University of Pavia, Via Taramelli 12, 27100 Pavia, Italy; daniele.merli@unipv.it (D.M.); giancarla.alberti@unipv.it (G.A.)

**Keywords:** 2-Furaldehyde (2-FAL), molecularly imprinted polymer (MIP), surface plasmon resonance (SPR), plastic optical fiber (POF), optical device, oil–paper insulation monitoring

## Abstract

2-Furaldehyde (2-FAL) is one of the main by-products of the degradation of hemicellulose, which is the solid material of the oil–paper insulating system of oil-filled transformers. For this reason, it has been suggested as a marker of the degradation of the insulating system; sensing devices for 2-FAL analysis in a wide concentration range are of high interest in these systems. An optical sensor system is proposed; this consists of a chemical chip, able to capture 2-FAL from the insulating oil, coupled with a surface plasmon resonance (SPR) probe, both realized on multimode plastic optical fibers (POFs). The SPR platform exploits gold nanofilm or, alternatively, a double layer of gold and silicon oxide to modulate the sensor sensitivity. The capturing chip is always based on the same molecularly imprinted polymer (MIP) as a receptor specific for 2-FAL. The system with the SPR probe based on a gold nanolayer had a higher sensitivity and a lower detection limit of fractions of μg L^−1^. Instead, the SPR probe, based on a double layer (gold and silicon oxide), has a lower sensitivity with a worse detection limit, and it is suitable for the detection of 2-FAL at concentrations of 0.01–1 mg L^−1^.

## 1. Introduction

Furfural (furan-2-carbaldehyde, or 2-Furaldehyde, or 2-FAL) is a furanic compound that is ubiquitous in environmental and food contexts, deriving from different chemical processes such as the degradation of sugars and cellulosic compounds, mainly by the action of heat and/or by the Maillard reaction [1,2,3,4]. It is also a promising renewable platform molecule derived from hemicellulose and related furanic compounds, in particular, hydroxymethyl-furfural (HMF) [5]. It is widely present in foods such as honey [6], in wine and alcoholic beverages [7], and in milk and its derivatives [8], where it is often analyzed as a quality marker, despite (due to its possible adverse) effects on human health [9]. Considering non-food matrices, it can be found in transformer oils due to the degradation of paper used as a solid insulator [10,11], and in this context, its determination is mandatory to monitor the degradation processes of solid insulating systems. Kraft paper or Thermally Upgraded Kraft paper is commonly used [12]; the latter is an alternative to the classical Kraft paper when high operating temperatures are expected [12]. Nevertheless, Kraft paper–oil insulation is still widely present as the insulating system of windings in many low and medium-voltage transformers in distribution networks, and high furans concentrations could be expected from its degradation so that a 2-FAL concentration in transformer oils ranging from 0.1 mg L^−1^ up to 5 mg L^−1^ can be expected [13].

The more widely used analytical methods for the determination of contaminants in many different fields are the chromatographic ones; however, the sensing methods for measurements outside the laboratory are of large interest also because of their low cost, in situ applicability and fast analytical response [14,15,16,17]. These characteristics are well reflected in a particular kind of surface plasmonic resonance (SPR) sensor based on multimode plastic optical fibers (POFs) [18,19,20,21,22] and using molecularly imprinted polymers as receptors [23,24]. In the case of 2-FAL determination, an SPR sensor based on a side polished POF and using molecularly imprinted polymers (MIPs) as receptors has been successfully developed and applied to matrices like water [25], wine [26] and even oil, mineral or vegetal [27]. The sensing platform consists of a D-shaped plastic optical fiber (POF), over which layers of buffer, gold, and MIP are successively deposited. They presented a detectable concentration range for 2-FAL of 0.001–0.1 mg L^−1^ [27], which is satisfactory for testing Kraft paper–mineral oil insulating systems but is not useful in the case of systems composed of stabilized Kraft paper, for which much lower concentrations can be indicative of degradation [28]. Thus, it is mandatory to develop sensing devices with a detection limit lower than that obtained with SPR-POFs.

Recently, a new sensing device, based on SPR and a MIP as the receptor, has been proposed for 2-FAL detection in water, obtaining ultra-low detection limits [29]. In this sensing device, the capturing zone, where the MIP is located, and the SPR surface are placed in different chips, both built on POFs. In the capturing chip, the MIP is placed in one or more micro-holes drilled in the core of the POF (POF-MIP). Thus, when the analyte (2-FAL) binds to the specific sites present in the MIP, the refractive index (RI) within the POF core is changed, and this produces a variation in the modal profile of the guided light, which is launched toward the second chip, which is an SPR-POF probe. This plasmonic probe is the same type as the SPR platform, which is in direct contact with the receptor in previously proposed sensors [25,26]. A change in the plasmonic resonance is produced by the variation in the modes launched in the D-shaped sensing region of the POF probe; this is, in turn, due to the variation in the light modes inside the capturing chip due to the adsorption of the target molecule. The RI of the dielectric in contact with the SPR probe surface is kept constant [29].

Very low detection limits for 2-FAL in water have been obtained by these sensors (0.04 µg L^−1^), even if the MIP was the same as previously used for the conventional SPR sensor [29]. The very high sensitivity of the double-chip device made it possible to take advantage of the presence of sites with high affinity (*K*_aff_ = 3025 L mg^−1^), able to bind 2-FAL even at very low concentrations, i.e., fractions of nM. It has been shown that the sensitivity (and the LOD) depends not only on the affinity of the receptor for the target molecule, but also on the structure of the chemical chip. Here, this structure was slightly modified with the aim of improving the detection limit.

Moreover, another modification of the original sensing system used in water [28] has been investigated, i.e., the variation in the dielectric’s RI over the gold film of the SPR probe. When the RI of the medium in contact with the SPR surface changes, the SPR condition and, therefore, the shape and wavelength of the SPR dip are expected to change accordingly [30,31].

Here, a nanolayer of silicon oxide (SiO_x_), which has an RI of 1.49 in the visible range, much higher than that of water, has been deposited over gold in the D-shaped POF SPR region.

Therefore, the same capturing chip is expected to produce different dose–response curves with a different sensitivity. This can be exploited to tune the detection range of the sensing device. A high sensor sensitivity can be achieved when the fixed RI on the SPR area surface is low, for example, 1.332, i.e., that of water [29].

## 2. Sensing Principle and Sensitivity Tuning

An innovative sensing strategy for bio/chemical applications has been recently developed by Cennamo et al. [29,32], exploiting the SPR phenomenon in a different way with respect to the more widely used SPR methodology in which a receptor layer is deposited on the plasmonic surface [26,27]. In the new strategy, on the contrary, a MIP in a bulky form is inserted as a receptor into micro-incisions in the core of the multimode waveguide, that is, in the core of the plastic optical fiber (POF). If the MIP has an RI higher than that of the core of the POF fiber (polymethyl methacrylate, PMMA), it is crossed by the light, behaving as the fiber’s core. This sensitive chemical waveguide is located before the SPR platform, which is used as the detector. Therefore, the whole sensing device is based on a multimode waveguide (the MIP-modified POF capturing chip) in which the modal profile of the propagating light changes when the binding of the target molecule with the MIP occurs. Therefore, a shift in the wavelength of the SPR resonance at the SPR probe occurs.

Figure 1 shows an outline of the sensing principle, which is similar to the one reported in previous work [32] by the same authors.

When the RI of the MIP filling the capturing chip changes because of the binding of the target molecule, a change in the propagated modes occurs [32]. For example, Figure 1 reports an alteration in the angles α_j_ and β_j_ representative of two propagated modes, when the template concentration in MIP changes.

The capturing chip, where the MIP is located, is connected in series with the SPR probe, and the variation in the light modes affects the SPR resonance. More specifically, when the RI of the dielectric in contact with the SPR surface increases, the higher modes increase their contribution to the convolution of different resonance peaks, and the dip is formed at higher wavelengths [31]. It must be underlined that this unconventional use of the SPR phenomena for bio/chemical sensing makes it possible to separate the sensing capturing chip, where the MIP has been inserted, from the SPR probe [29]. Moreover, it is well established that the measured SPR dip in the SPR platform, which is based, in turn, on a multimode waveguide (modified POF), is the result of a convolution of different resonance dips (one per each propagation mode) and that each dip corresponds to a specific resonance condition satisfied by a given angle–wavelength coupling [30]. Therefore, when the RI of the medium in contact with the SPR surface changes, the resonance condition changes, and both the shape and wavelength of the SPR dip change [31]. On this basis, the sensitive capturing chip, connected in series with the SPR platform, can have different effects on the variation in the SPR dip if the RI of the medium in contact with the SPR area is different. More specifically, when the RI in contact with the SPR surface increases, the higher modes increase their contribution to the convolution of different resonance dips.

Considering that the MIP’s RI variation and any other change in the core of the sensitive multimode chip heavily affect the propagation of the fundamental mode and lower modes, better sensor sensitivity can be achieved when the fixed RI on the SPR area is low, for example, 1.332, i.e., that of water [32]. On the other hand, by changing the value of the fixed RI in contact with the SPR region, the same capturing bio/chemical chip (where the MIP is located) can produce responses with a different sensitivity. This principle is exploited to tune the device’s detection range by changing the RI of the medium in contact with the SPR surface.

## 3. Materials and Methods

### 3.1. Chemicals

The target molecule 2-Furaldehyde, 2-FAL (CAS No. 98-01-1), the functional monomer methacrylic acid (MAA, CAS No. 79-41-4), the cross-linker divinylbenzene (DVB, CAS No. 1321-74-0), and the radical initiator 2,2′-azobisisobutyronitrile (AIBN, CAS No. 78-67-1), all of analytical reagent grade, were obtained from Sigma-Aldrich (St. Louis, MO, USA), and, when required, were purified by SPE on an alumina column immediately before utilization.

The transformer-insulating mineral oil was an uninhibited naphthenic mineral oil, which is typically used as an insulating oil in transformers. It was provided by Nynas Corp. (Stockholm, Sweden) and certified as not containing any furanic derivative.

The oil solutions of 2-FAL were prepared by dissolving the concentrated standard in the oil. The lower concentrations were obtained by successive dilution.

### 3.2. Preparation of the Molecularly Imprinted Polymer

As reported before, the MIP prepolymer mixture was prepared using a non-covalent strategy for sensing oil samples and the conventional SPR sensors SPR-POF, in which the MIP is in direct contact with the resonant surface [27]. The same pre-polymeric mixture was also used for sensing 2-FAL in aqueous matrices either by SPR-POF with MIP [25] or using the capturing chips coupled with the SPR probe [33].

The molar ratio of the components 2-FAL/MAA/DVB was 1/4/40. Briefly, MAA was uniformly dispersed in DVB by sonication; then, 2-FAL was added, and the solution was reaerated for 10 min with nitrogen. Finally, AIBN was added to the liquid mixture at about 20 mg/mL. Divinylbenzene (DVB) was used in large excess so that it was, at the same time, the cross-linker and the solvent. No other solvent was used in the pre-polymeric mixture to obtain a structure as rigid as possible.

### 3.3. Preparation of the SPR Probe

The SPR detector (named SPR) is an SPR platform that has been widely used in previous applications, as reported for example in [27]. It consists of a 10 mm long plastic optical fiber (POF; 980 μm core, 10 μm cladding) where the cladding and half of the core have been removed by lapping and polishing procedures. Previously, the POF was fixed on a resin holder of 10 × 10 × 10 mm. On the planar surface, a photoresist buffer layer (thickness: 1500 nm) and a gold nanofilm (60 nm thick) are deposited on the exposed core. The metal film is deposited by a sputtering machine (Bal-Tec SCD 500) (Gemini B.V., Apeldoorn, The Netherlands), as reported in [34], and it is in contact with the external dielectric medium (Figure 2a). As previously found in similar devices, double-distilled water is particularly suited to this process because of its RI of 1.332 refractive index units (RIUs).

A second kind of SPR platform (named SPR-SiO_x_) was obtained by evaporating an overlayer of silicon oxide on the gold plasmonic nanofilm. The thin oxide layer (40 nm) was evaporated in an evaporator BAK640 Evatec under oxygen partial pressure (below 3 × 10^−4^ bar), while the deposition rate (0.5 nm/s) was monitored by a quartz crystal system. This type of SPR platform was developed to excite an SPR resonance at a longer wavelength with respect to the previous SPR platform configuration at the interface, with double-distilled water used as an external medium (Figure 2b).

### 3.4. Preparation of the Micro Trench-Based Capturing Chip with MIP

The capturing chip (named POF-MIP_2-FAL_) (Figure 3) is made from a POF (980 μm core, 10 μm cladding) modified to eliminate the cladding and to obtain a D-shaped profile by a lapping and polishing procedure, as in the case of the SPR probe described above [27,33]. A micro-trench (incision 6 mm long, 600 µm wide and 600 µm deep) is drilled into the flat part of the plastic optical fiber using a computer numerical control (CNC) micro-mill (Figure 3). The micro-trench is carved in an orthogonal direction with respect to the light propagation along the fiber.

The POF chip is then filled with a drop (10 μL) of the MIP pre-polymeric mixture and maintained at 80 °C for 16 h in an oven for thermal polymerization [27]. The template is extracted from the MIP by repeated washing with 96% ethanol and finally with distilled water. The remaining MIP appears as a straight line lying on the exposed core of the fiber, as shown in the picture in Figure 2b.

A capturing POF chip with a micro-trench was preferred to other layouts, like the micro-hole structured POF [29,33], for chemical sensing in insulating oil. This layout guarantees a large area of contact between the MIP and the drop of insulating oil deposited on the MIP surface. In fact, due to the highly lipophilic nature of the MIP, the oil drop tends to expand over the MIP surface, enabling a large area of interaction between the synthetic receptor and the chemical marker (2-FAL) dissolved in insulating oil. Moreover, the core modification is larger, possibly giving a better sensitivity.

### 3.5. Modular Testing Unit

A modular testing unit has been designed and prepared using 3D printing-based prototyping (Figure 4).

It includes two separate modules: one with a capturing unit (a), which hosts the MIP-functionalized micro-structured POF (POF-MIP_2-FAL_), and the detecting unit (b), consisting of an SPR-POF platform (SPR or SPR-SiO_x_) and a spectrometer with a detection range of 350 nm to 1000 nm (FLAME-S manufactured by Ocean Insight, Orlando, FL, USA). The capturing unit is fed by a halogen lamp with a spectral emission range of 360 nm to 1700 nm) that is connected to the external light source via an optical POF cable (1 m long). The whole testing devices are named SPR/POF-MIP_2-FAL_ and SPR-SiO_x_/POF-MIP_2-FAL_.

### 3.6. Measurement Procedure

The measurement of the 2-FAL concentrations with the assembled sensing device is performed via the following steps:-Reference spectrum acquisition: the reference spectrum is acquired with a drop of pristine liquid (water, insulating oil) over the capturing chip in contact with the MIP surface and air over the SPR-POF probe.-Blank transmitted spectrum acquisition: the SPR transmitted spectrum is acquired with pristine liquid (e.g., fresh oil) over the capturing chip and water over the SPR probe. The acquired spectrum is normalized to the reference spectrum.-Dose-response curve based on SPR spectra. The SPR transmitted spectra are acquired with water over the SPR probe surface and pristine liquid (e.g., fresh oil) over the capturing chip, dropped after a 10 min incubation with samples of increasing 2-FAL concentrations in contact with the MIP surface (the capturing chip) and a washing step with pristine liquid (e.g., fresh oil) to remove the molecules not specifically adsorbed. These acquired spectra are normalized to the reference spectrum to obtain the SPR spectra.

All the tests were conducted in a laboratory-controlled environment to prevent water evaporation from the SPR surface during measurements.

## 4. Results

The SPR detection platform was the same as that previously used for the SPR sensors in which the receptor, i.e., the specific MIP, was directly in contact with the metal nanolayer. This kind of SPR platform has been previously proposed for the detection of many analytes, including 2-FAL, in aqueous solution [25] and in non-aqueous media such as insulating oil [27]. Unlike previous applications, in the sensing device considered here, the gold layer is in contact with a medium at a fixed RI that is kept constant during the entire measurement procedure, so that the SPR variation is only due to the variation in the modes of the light in the fiber coming from the micro-trench MIP chip. The SPR probe is used with water as a medium above the resonant layer because the RI of water (1.332) produces a resonance at the suitable wavelength of around 600 nm. Another kind of SPR platform to be used as a detector has been investigated here. This platform has a silicon oxide (SiO_x_) layer over the gold nanofilm with a 40 nm thickness, which assures a uniformly distributed dielectric layer with a high RI. The probe is indicated as SPR-SiO_x_; in this case, water was also used as the medium over the SiO_x_ layer.

### 4.1. Transmission Spectra of the SPR Probe

The two families of the SPR platforms used as detectors, one without (SPR probe) and one with the silicon oxide layer over the gold nanofilm (SPR-SiO_x_ probe), have been characterized by recording their spectra in water. Some transmission spectra are reported in Figure 5, where a significant difference between the spectra of the two kinds of platforms is noticed, which confirms that the SiO_x_ layer has been successfully deposited.

The difference is even better appreciated in the normalized spectra, as reported in Figure 6. Figure 6a,b show the spectra in water normalized on the spectra of the same platform in air, registered immediately before the measurement in water, respectively, for the SPR-POF and SPR-SiO_x_ probes. Here, the normalized spectra of different SPR platforms (named s0–s4) are reported in order to evaluate their reproducibility. It must be noticed that the normalized spectra in Figure 6a, relative to the SPR probes with only the gold layer, have an SPR resonance at about 600 nm (mean value 601 ± 2 nm), in agreement with previous observations [27].

The normalized spectra in Figure 6b, related to SPR-SiO_x_ probes s5–s7, show an SPR minimum at about 700 nm (mean value 692 ± 3 nm); thus, a shift of about 100 nm has been produced by the presence of the thin SiO_x_ overlayer with water on top. The irreproducibility of the dips’ minima is similar in the two kinds of probes and not higher than that of the SPR-POF probes with MIP in contact with the gold layer [27]. The SiO_x_ layer does not add any further irreproducibility to the sensing device. Notice also that in these probes, a peak at about 580 nm is evident, possibly due to a lossy mode resonance phenomenon (LMR) [35].

### 4.2. Characterization in Water of the Micro-Trench-Based SPR POF-MIP_2FAL_ Sensing System

For the sake of comparison with a similar device previously proposed, based on the same MIP but with a differently structured POF chip [29], the performance of the new sensing system, based on a micro-trench structured chip coupled with the SPR probes (SPR POF-MIP_2-FAL_), in detecting the 2-FAL concentration in water solution was tested. The SPR detector was an SPR probe with water (1.332 RIU) upon the gold surface. The SPR spectra for the dose–response curve were acquired by placing a drop of aqueous sample solution, with different concentrations of 2-FAL, over the MIP platform; these were then normalized over the reference spectrum previously acquired (Figure 7a). They show an SPR dip at about 610 nm, which shifts toward lower wavelengths (blue-shift) when the 2-FAL concentration in the sample increases. The dose–response curves, obtained by plotting the resonance wavelength shifts (∆λ, nm) vs. 2-FAL concentration (ranging from 0 mg L^−1^ to 5 mg L^−1^), are reported in Figure 7b. For concentrations higher than 0.01 mg L^−1^, a constant signal is reached.

The Langmuir isotherm equation (Equation (1)) (full line in the graph) was used to fit the experimental data; this model is generally used to reproduce the MIP-based sensing response according to the Langmuir isotherm equation [26,27]:(1)$$\Delta \lambda =\frac{\Delta \lambda_{\max}\cdot K_{\mathrm{aff}}\cdot c_{2-\mathrm{FAL}}}{1+K_{\mathrm{aff}}\cdot c_{2-\mathrm{FAL}}}$$
where ∆λ_max_ is the maximum value of the wavelength shift ∆λ (nm) at the plateau of the curve, *K*_aff_ (L mg^−1^) is the affinity constant of the adsorption equilibrium, and *c*_2-FAL_ (mg L^−1^) is the 2-FAL concentration. Table 1 summarizes the parameters obtained from the non-linear fitting; the comparison with another capturing MIP-based chip (MIP-POF1, i.e., a chip with the MIP located in a cylindrical micro-hole [29]) is also reported.

As shown in Table 1, the affinity constants obtained with both chips are similar, as expected, since the same MIP_2-FAL_ is used. Comparable sensitivity values at low 2-FAL concentrations (i.e., the slope of the linear part of the dose–response curve) and LOD (computed as $3.3\cdot s_{y/x}/\mathrm{slope}\;\mathrm{at}\;\mathrm{low}\;c_{2-\mathrm{FAL}}$, where *s_y_*_/*x*_ is the standard error of the estimate, obtained from the linear regression of the data; this value can be assumed to be not significantly different from the standard deviation of the replicate measurements of the blank solutions [36]) were obtained too.

The Δλ_max_ is higher in the case of the micro-trench considered here, and it is probably due to the larger amount of MIP in the trench in the core of the plastic fiber and the longer light path.

The optical signal reaches a plateau at a 2-FAL concentration of about 0.01 mg L^−1^, like the trend observed with the MIP-POF1, with one micro-hole previously studied [29]. On the contrary, the results of the here-developed platform differ from those obtained with a similar device in water [29] but with three micro-holes instead of only one. In that case, two kinds of sites for 2-FAL were detected, the second being weaker. This made it possible to detect a wider concentration range, about four orders of magnitude instead of only two.

### 4.3. Characterization of SPR/POF-MIP_2-FAL_ Sensing System in Insulating Oil

The sensing system based on the SPR detector and micro-trench-based chip filled with MIP specific for 2-FAL (MIP_2-FAL_) was used for the detection of 2-FAL in insulating oil. The SPR response to various sample solutions at different 2-FAL concentrations prepared in uninhibited naphthenic mineral oil was investigated. The same data treatment described in the previous paragraph was employed (see Equation (1)).

As an example, Figure 8a shows the SPR spectra in oil for different concentrations of 2-FAL, normalized to the reference spectrum; this last one had been previously recorded, with the SPR detector in air and the MIP surface of the chemical chip in contact with a drop of oil with a 2-FAL concentration equal to 0 used as a blank solution. The SPR minima at about 618 nm shifts toward lower wavelengths (blue shift) by increasing the concentration of the analyte. In Figure 8b, the corresponding dose–response curve is reported.

The ∆λ_max_ value in insulation oil is 3.4 (0.1) nm, like that obtained in water (see Table 1), while the *K*_aff_ in oil is 3897 (840) mg^−1^ L, somewhat higher than that in water. This difference agrees with the results obtained with classical SPR–MIP_2-FAL_ sensors in which the MIP is in direct contact with the SPR surface. This difference is reasonably due to the significantly different experimental conditions; it is, in fact, a conditional constant strongly dependent on the medium, temperature, pH, etc. Significantly higher is the sensitivity at a low concentration, at 13,143 nm mg L^−1^, and the low detection limit is about 0.00003 mg L^−1^, 1.6 times lower than the detection limit in water. This shows that even in oil, this sensor performs well at very low concentrations of 2-FAL (fractions of µg L^−1^), much lower than that obtained with the SPR platform where the MIP is in contact with the resonant surface. At the same time, the optical signal saturates at a 2-FAL concentration of about 0.01 mg L^−1^, as observed in aqueous medium. Therefore, the sensing system is not able to monitor higher 2-FAL concentrations.

The *K*_aff_ values in insulating the mineral oil determined by the SPR probe and the POF-MIP _2-FAL_ chip (SPR-POF-MIP _2-FAL_) are impressively higher than that obtained in the case of the sensor with MIP in contact with the resonant surface [27], *K*_aff_ = 45 L mg^−1^; this could indicate that at least two different kinds of MIP recognition sites exist, but only those with a high affinity for 2-FAL could be detected in the present investigation because of the higher sensitivity of the sensing method [25]. Significantly higher is the sensitivity at a low concentration, namely 13,143 nm mg^−1^ L; moreover, a slightly lower detection limit of about 0.03 µg L^−1^, 1.6 times lower than that obtained in water, is observed. These pieces of evidence show that in oil, too, this photonics layout performs well at very low concentrations of 2-FAL (fractions of µg L^−1^), much lower than that obtained with the SPR platform in which the MIP is in contact with the resonant surface [27]. The optical signal reaches a plateau at a similar 2-FAL concentration, as observed in an aqueous medium. Therefore, this sensing system is unsuitable for monitoring analyte concentrations in oil higher than 0.01 mg L^−1^.

### 4.4. Response in Oil of the SPR Platform with SiO_x_ Overlayer Coupled with a MIP_2-FAL_-Filled Micro-Trench Chip (SPR-SiO_x_/POF-MIP_2-FAL_)

The capturing chip with a micro-trench filled with MIP (POF-MIP_2-FAL_) has been used with the second type of SPR probe, i.e., that with the silicon oxide thin film evaporated on the gold nanofilm, namely the SPR-SiO_x_ platform. The performance of the SPR-SiO_x_/POF-MIP_2-FAL_ sensing system in insulating oil with 2-FAL concentrations ranging from 0.02 mg L^−1^ up to 3 mg L^−1^ was assessed at about 707 nm, which is the resonance wavelength of the considered SPR-SiO_x_ probe, as seen in Figure 6b. The normalized SPR spectra recorded at different 2-FAL concentrations are shown in Figure 9. Notice that the SPR resonance wavelength shifts toward longer wavelengths when increasing the 2-FAL concentrations (red shift). This is due to the different impacts of the modes (higher modes instead of the lower modes) in the resonance convolution in the SPR phenomena. A similar shift in the LMR peak at about 590 nm is evidenced in the same figure [35].

Figure 9b shows the dose–response curve for ∆λ (nm) vs. the 2-FAL concentration, averaged on three chemical chips (POF-MIP_2-FAL_) prepared using the same procedure and exposed to 2-FAL standards in mineral oil.

The parameters of the dose–response curve obtained by the Langmuir fitting according to Equation (1) are reported in Table 2.

In the case of the SPR-SiO_x_/POF-MIP_2-FAL_ sensing system, the affinity constant and the sensitivity are some magnitudes lower than those obtained for the SPR/POF-MIP_2-FAL_ system and are more like the values obtained with the more used platform with the receptor in direct contact with the resonant surface [27]. Notice also that the *K*_aff_ values are like those detected in the case of the sensor with a three-hole chip in [29]. This confirms that two kinds of sites are present in the MIP_2-FAL_ considered, each of which is detected when the probe employed has a suitable sensitivity.

The SPR-SiO_x_ probe coupled with the SPR/POF-MIP _2-FAL_ capture chip, which responds to high 2-FAL concentrations, makes it possible to detect 2-FAL in a concentration range that could also be of interest for checking the health conditions of transformers, in particular those based on the Kraft paper that is more easily degraded. This could be due to the lower sensitivity of the device at 700 nm. The LOD is equal to 14 μg L^−1^, well above that obtained by operating at about 600 nm with the SPR-POF-MIP_2FAL_ sensor. Similar results in terms of the *K*_aff_ and LOD can be obtained by exploiting the LMR phenomenon; therefore, in a way similar to [35], it is possible to detect 2-FAL concentrations via SPR or LMR phenomena. Some recovery experiments carried out at two concentration levels in mineral oil, i.e., 1 μg L^−1^ and 100 μg L^−1^, respectively, with SPR/POF-MIP_2FAL_ and SPR-SiO_x_/POF-MIP_2FAL_ gave a recovery of 119% and 115%, which can be considered satisfactory. The sample with 1 μg L^−1^ 2-FAL did not give any signal in the SPR-SiO_x_/POF-MIP_2FAL_ device, while the same sample at SPR/POF-MIP_2FAL_ gave ∆λ = 3.4 nm, corresponding to the plateau value. Neither of the sensing devices was able to give any useful signal for concentrations around 10 μg L^−1^ 2-FAL.

To further investigate the response of the considered sensing system, a different SPR-SiO_x_ platform was considered. Here, the gold layer was deposited by evaporation by a thermal evaporator, BAK640, Evatec, instead of by sputtering. This could produce a gold surface to which the silicon oxide could adhere in a different way to that on the gold surface obtained by sputtering. The response to 2-FAL at different concentrations in mineral oil was investigated by recording the SPR spectra at different concentrations of 2-FAL. The spectra show a dip minimum at about 680 nm, a lower wavelength than in the case of the other SPR-SiO_x_-/POF-MIP _2-FAL_ systems examined here. The dose–response curve is reported in Figure 10.

At concentrations higher than 0.1 mg L^−1^, it is similar to the dose–response curve obtained in the case of the SPR-SiOx/POF-MIP 2-FAL sensing system with sputtered gold, while at concentrations lower than about 0.0001 mg L^−1^, the response is similar to that of the SPR/POF-MIP _2-FAL_ device. This could be ascribed to only a partial covering of the gold surface by the thin layer of SiO_x_.

The observed double behavior can be justified by considering the presence of two types of interaction sites with different affinities. For low concentrations of 2-FAL, the higher affinity MIP combination sites will be occupied first, and this is monitored by the part of the sensing probe with gold only; as the concentration is increased, the lower affinity sites will also be occupied [25], and this is monitored by the part of the sensing probe with SiO_2_ over gold.

The double behavior of the standardization curve in Figure 10 was fitted, similarly to the fitting procedure adopted in previous works for a chemical chip with three micro-holes [29], using the following equation:(2)$$\Delta \lambda =\frac{\Delta \lambda_{\max,1}\cdot K_{\mathrm{aff},1}\cdot c_{2-\mathrm{FAL}}}{1+K_{\mathrm{aff},1}\cdot c_{2-\mathrm{FAL}}}+\frac{\Delta \lambda_{\max,2}\cdot K_{\mathrm{aff},2}\cdot c_{2-\mathrm{FAL}}}{1+K_{\mathrm{aff},2}\cdot c_{2-\mathrm{FAL}}}$$
where *K*_aff1_ is the affinity constant for sites with higher affinity and *K*_aff2_ is the affinity constant for those with lower affinity. The *K*_aff_ values obtained from the fitting of the dose–response curve in Figure 10 are *K*_aff1_ = 8130 (2720) mg L^−1^ and *K*_aff2_ = 13 (9.4) mg L^−1^, respectively. Meanwhile, with a high uncertainty, it can be seen that the affinity constants evaluated are similar to those of SPR/POF-MIP_2-FAL_ for the strong sites (*K*_aff1_) and SPR-SiO_x_/POF-MIP_2-FAL_ (*K*_aff2_) for the weaker sites, respectively, evaluated above and reported in Table 1 and Table 2.

It seems, therefore, possible to tune the sensitivity range of the sensor system by acting on the design parameters of the chemical chip, of the SPR probe chemical chip or of the SPR probe.

Notice that this sensing device allows for the quantification of the 2-FAL concentration in the ultrawide concentration range from 0.0001 to 10 mg L^−1^. At a 10 µg L^−1^ concentration, which could not be evaluated by the devices described above, the recovery was 85%, as obtained graphically from the dose–response curve.

## 5. Discussion and Conclusions

The sensing device with the SPR probe and the capturing chip with MIP_2-FAL_ (SPR/POF-MIP_2-FAL_), with the gold layer in contact with water, can be used for the determination of very low concentrations of 2-FAL in mineral oil, in a concentration range of 10^−4^ to 10^−2^ mg L^−1^ with a LOD of 46 ng L^−1^ (0.5 nM). The resonance wavelength is around 600 nm, with a blue shift at increasing concentrations of 2-FAL. Concentrations higher than 0.01 mg L^−1^ cannot be determined since there is not any variation in response when the concentration is increased, despite the fact that other kinds of sites, with lower affinity, are present in the considered MIP; this has been found by other previously described SPR sensing methods, and also by the other kinds of sensing system (SPR-SiO_x_/POF-MIP_2-FAL_) examined in the present investigation. Indeed, concentrations of 2-FAL from 0.01 to 1 mg L^−1^ can be determined by using the SPR-SiO_x_/POF-MIP_2-FAL_ sensor, with a layer of silicon oxide over gold, for which the resonance wavelength is about 700 nm. Using this kind of sensing system, 2-FAL concentrations in the range of 10^−4^–10^−2^ mg L^−1^ do not produce any wavelength variation because of the sensor’s low sensitivity.

The combination of 2-FAL with the weaker sites does not produce any signal that is detectable by the sensing system working at about 600 nm, SPR/POF-MIP _2-FAL_. A possible explanation could be the different spatial distribution of the two kinds of MIP sites in the micro-trench and the different influence of the light modes (higher or lower modes) in the resonance convolution in the SPR phenomena.

It must be underlined that binding site heterogeneity is a well-known phenomenon, particularly in the case of MIPs synthesized by the noncovalent strategy [37,38].

In view of the determination of the concentration in the unknown sample, it must be noticed that a dose–response curve that can be fitted considering only homogeneous binding sites, as in Equation (1), usually makes it possible to evaluate concentration ranges not larger than two orders of magnitude. This can be convenient from another point of view since, once the parameters are known, the following Equation (3) can be used for the evaluation of the unknown concentration of the sample
(3)$$c_{2-\mathrm{FAL}}=\frac{\Delta \lambda_{x}}{\left(\Delta \lambda_{\max}-\Delta \lambda_{x}\right)\cdot K_{\mathrm{aff}}}$$

A larger detection range can be obtained when more than one kind of MIP interaction site is involved in the response to the sensing interrogation, as in the case of the sensing system with sputtered gold proposed here. In this case, a more complicated relationship is required to evaluate the analyte concentration, as seen in Equation (2). It is important to highlight that the primary aim of the proposed method is to exploit SPR probes with different RI media for modulating the detectable concentration range of 2-FAL in the analysis of insulating oil for transformers. The method proved effective. Moreover, it has been found that the method of gold layer deposition also has an influence, making it possible to exploit the interaction sites of the MIP with different affinities and thus the analysis of ultrawide concentration ranges. The next step will be determining the analytical parameters of devices based on this principle, such as their accuracy, repeatability and reproducibility, which would be very interesting for practical applications.

## Figures and Tables

**Figure 1 sensors-24-05261-f001:**
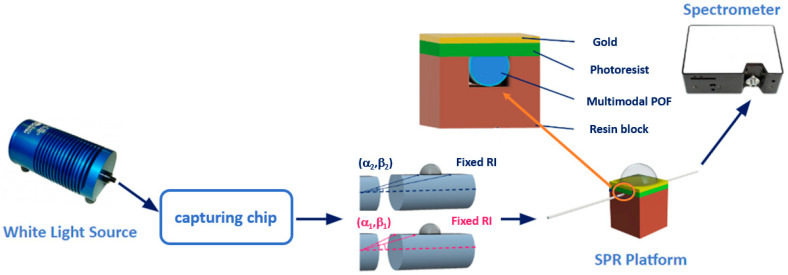
Outline of the sensing principle based on a modified POF filled with the MIP receptor and an SPR-POF platform: (α_1_,β_1_) and (α_2,_β_2_) are the angles representative of the two propagated modes before and after the MIP–analyte binding.

**Figure 2 sensors-24-05261-f002:**
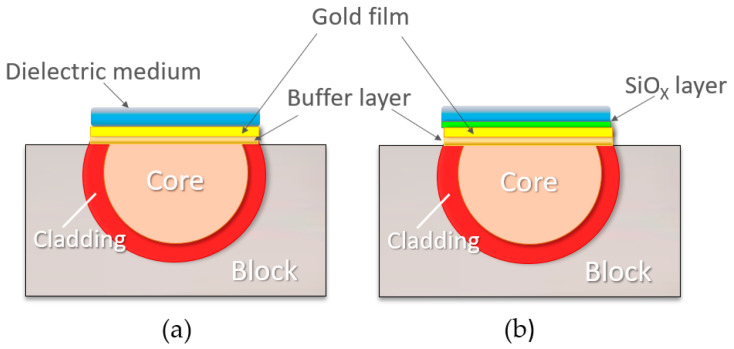
SPR platform: schematic side view of the D-shaped plastic optic fiber. The multilayer in contact with the dielectric medium is (**a**) a photoresist buffer layer (1500 nm thick) and a gold nanofilm (60 nm thick), and (**b**) a photoresist buffer layer (1500 nm thick), a gold thin film (60 nm thick) and a silicon oxide overlayer (40 nm thick).

**Figure 3 sensors-24-05261-f003:**
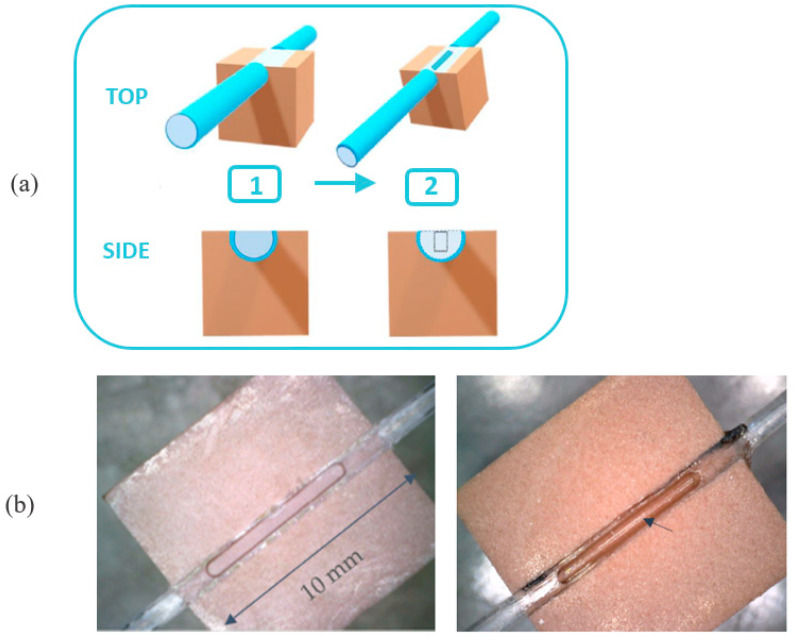
Capturing chip. (**a**) scheme of the plastic optical fiber (POF, 0.98 mm diameter) modified chip, inserted in the resin block; (1) picture of 10 mm D-shaped region of the POF [33]; (2) micro-trench (6 mm length, 0.6 mm depth, 0.6 mm width) in the core of the D-shaped POF. (**b**) Pictures of a micro-trench carved in the POF’s core (to the left) without MIP_2-FAL_ and (to the right) with MIP_2-FAL_.

**Figure 4 sensors-24-05261-f004:**
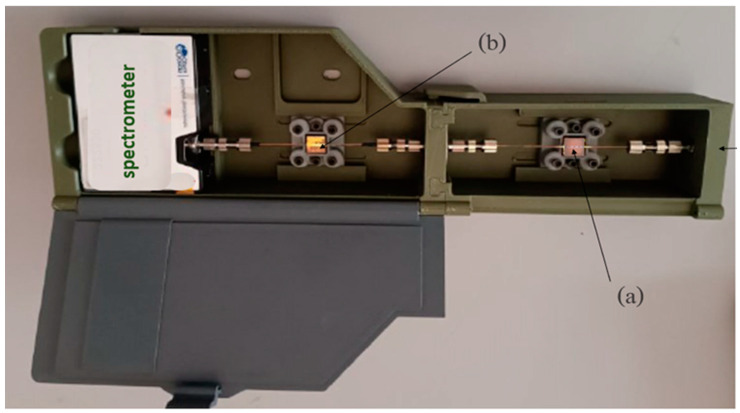
Testing unit with two interlocked modules: (**a**) chemical sensing unit (POF-MIP_2-FAL_) and (**b**) SPR detection unit with SPR platform (SPR or SPR-SiO_x_).

**Figure 5 sensors-24-05261-f005:**
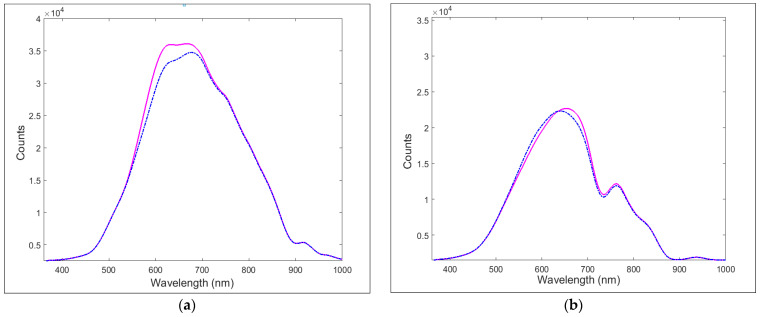
Transmission spectra of (**a**) SPR probe and (**b**) SPR-SiO_x_ probe, with air (pink solid line) and water (blue dotted line) as the dielectric over the SPR surface.

**Figure 6 sensors-24-05261-f006:**
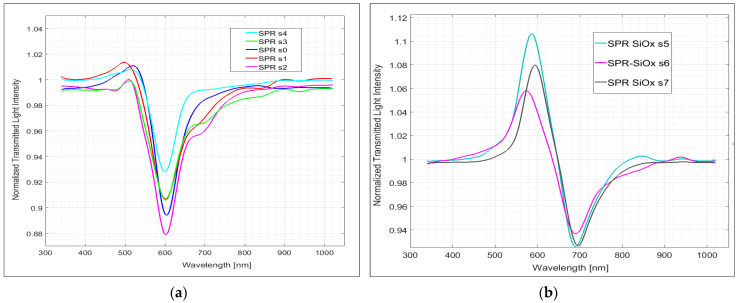
SPR spectra in water, normalized to the reference spectrum in air of different SPR sensors, named s0–s7: (**a**) SPR-POF probes with a photoresist and gold nanofilm: sensors s0–s4; (**b**) SPR-SiO_x_-POF probes, named s5–s7, with a photoresist, gold nanofilm (60 nm) and a silicon oxide (SiO_x_, 40 nm) overlayer.

**Figure 7 sensors-24-05261-f007:**
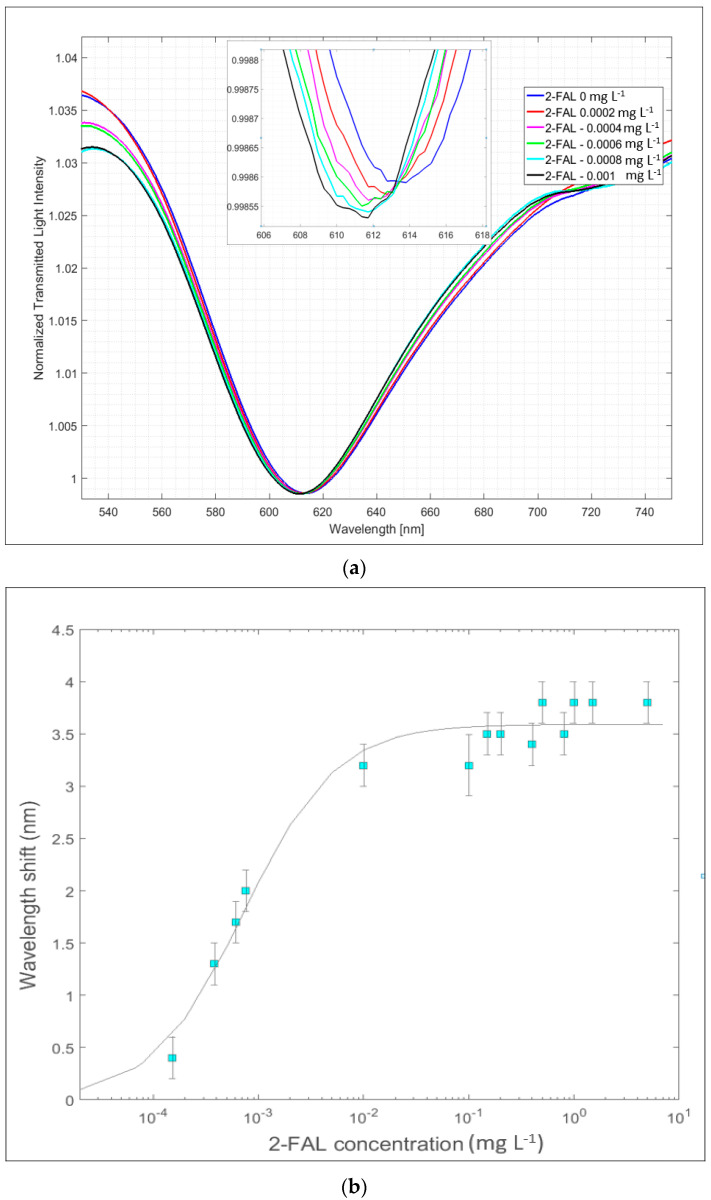
SPR/POF-MIP_2-FAL_ sensing system in water: (**a**) normalized SPR spectra at lowest 2-FAL concentration (0–0.001 mg L^−1^). (**b**) Absolute value of the shift in the resonance wavelength versus 2-FAL concentration (0–5 mg L^−1^), with the Langmuir fitting of the experimental data and error bars, in semi-log scale.

**Figure 8 sensors-24-05261-f008:**
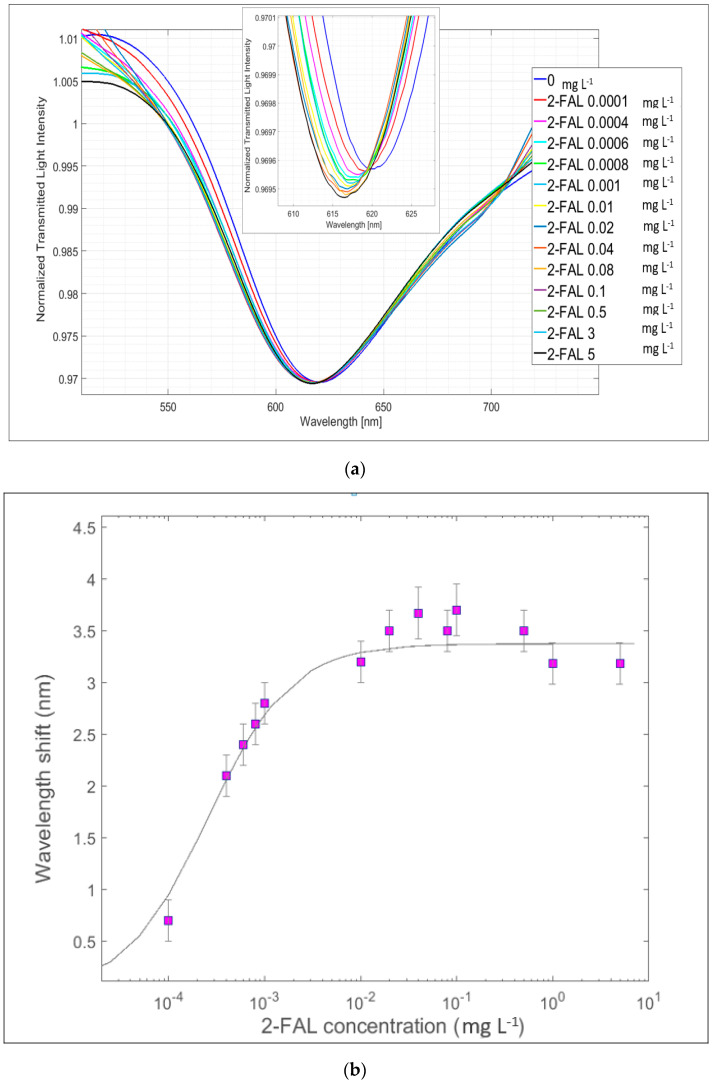
SPR response of the sensing device based on a SPR platform coupled with a MIP _2-FAL_-filled micro-trench chip in mineral oil (SPR/POF-MIP_2-FAL_)): (**a**) SPR spectra, at different concentrations of 2-FAL in oil, normalized on the blank spectrum, obtained by the exposition of the surface of the gold layer of the SPR platform to air and of the chemical chip in pristine insulating oil; (**b**) dose–response curves for 2-FAL in mineral oil.

**Figure 9 sensors-24-05261-f009:**
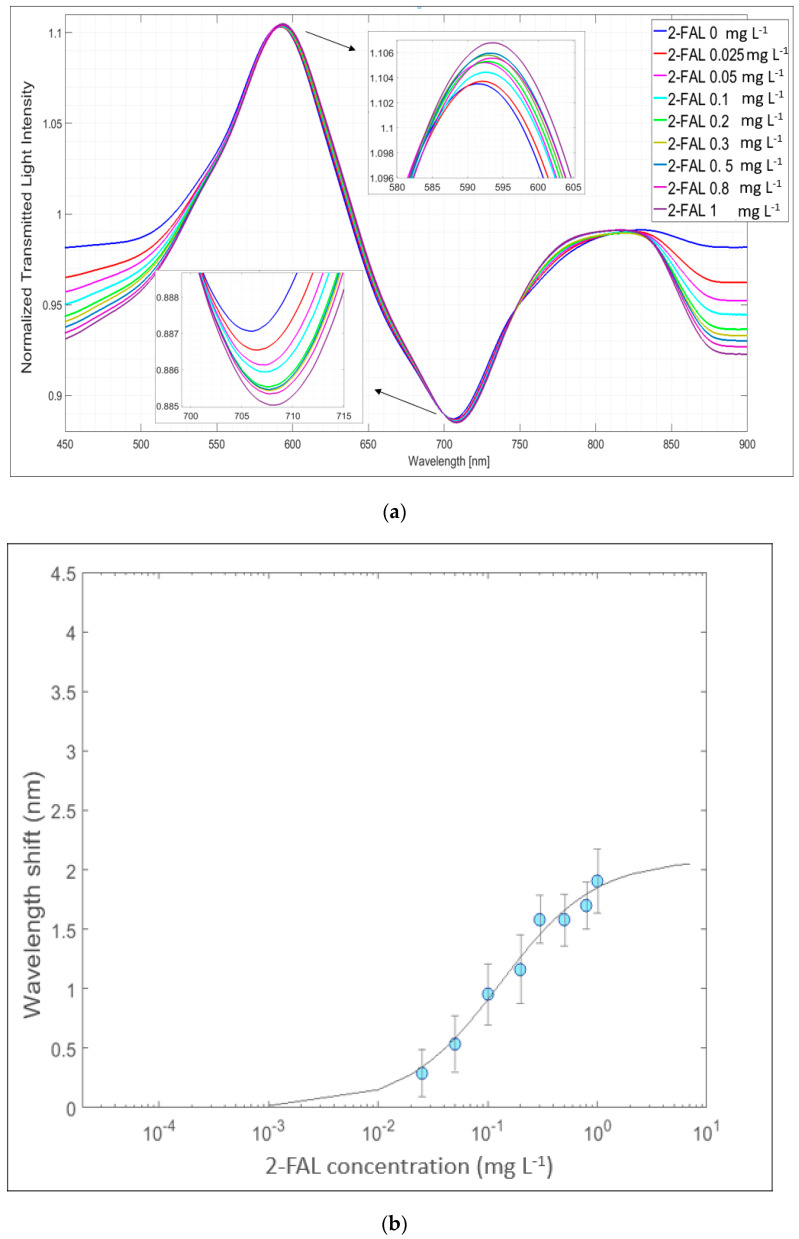
SPR-SiO_x_/POF-MIP _2-FAL_ sensing device with a probe with a gold nanofilm and silicon oxide film and a MIP _2-FAL_-filled micro-trench chip: (**a**) SPR spectra, at different concentrations of 2-FAL in oil, normalized on the reference spectrum; (**b**) Dose-response curve: ∆λ (nm) vs. 2-FAL concentration (logarithm scale) in mineral oil.

**Figure 10 sensors-24-05261-f010:**
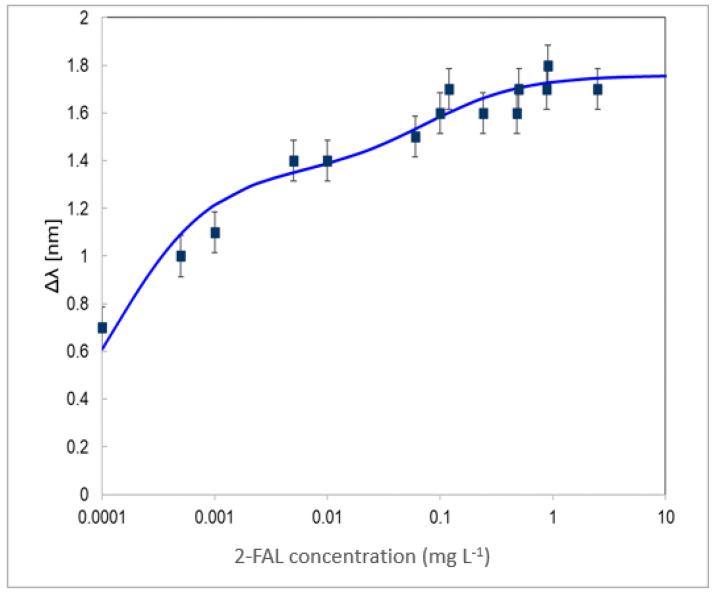
Dose–response curve of the SPR-SiO_x_/POF-MIP_2-FAL_ sensing device with a different gold layer. The solid curve is the fitting curve obtained by Equation (2). Experimental data (black); fitting (solid line).

**Table 1 sensors-24-05261-t001:** Parameters of 2-FAL in water sensors with SPR probe coupled with different capturing chips, based on the same MIP_2FAL_.

Sensing Device	Δλ_max_ nm (Std.Err.)	*K*_aff_ mg^−1^ L (Std.Err.)	Sensitivity at Low *c*_2-FAL_ (nm mg^−1^ L)	LOD (µg L^−1^)	R^2^
SPR-POF-MIP_2-FAL_ (micro-trench)	3.59 (0.07)	1371 (187)	4924	0.046	0.97
MIP-POF1 (one micro-hole) (a)	1.35 (0.06)	3025 (1030)	4084	0.048	0.95

(a)—[29].

**Table 2 sensors-24-05261-t002:** Parameters of standardization curves for 2-FAL in mineral oil for different SPR probes.

Sensing Device	Approximate λ_res_ nm	∆λ_max_ nm (Std.Err.)	*K*_aff_ mg^−1^ L (Std.Err.)	Sensitivity at Low *c*_2-FAL_ (nm mg^−1^ L)	LOD µg L^−1^	R^2^
SPR-POF-MIP_2-FAL_	600	3.4 (0.1)	3897 (840)	13,143 (0.2)	0.03	0.88
SPR-SiO_x_-POF-MIP_2-FAL_	700	2.0 (0.1)	13 (3)	27.0 (0.2)	14	0.93

## Data Availability

The data are available on reasonable request from the corresponding author.

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
