# Peer review of "An Optical Device Based on a Chemical Chip and Surface Plasmon Platform for 2-Furaldehyde Detection in Insulating Oil"

_sensors, 2024, doi:10.3390/s24165261_

Round 1
Reviewer 1 Report
Comments and Suggestions for Authors
In this manuscript, the authors proposed a SPR probe based optical sensor for 2-FAL detection in insulating oil samples. On the whole, this article is interesting and well-written. In my opinion, a minor revision needs to be made before acceptance.
-
Compared to other sensing methods for the determination of 2-FAL, What is the advantage of SPR sensor?
-
Figure 5, what is S0, S1, S2, …? Are they some parallel samples?
-
Line 278-285, 241-256, 228-238, what is the reason to create many very short subparagraphs?
-
What is the background level of 2-FAL in your sample? Have you ever evaluated the accuracy of the sensor by recovery experiments?
-
The specificity experiment is missing.
-
Some formatting issues are expected to be corrected. Just some examples below:
- line 3, oil. → oil; line 5, Pesavento4,+. → Pesavento4,+; line 13, Correspondence → + Correspondence
- line 26, mg l-1 → mg L-1; line 298, 5 mg l-1 → 5 mg L-1
- line 74, plasmonic resonance (SPR) → SPR; line 141, 2-furaldehyde (2-FAL, CAS N. 98–01–1) → 2-FAL (CAS No. 98-01-1). All the abbreviations should be stated when the corresponding word first appears in the manuscript and always use the abbreviations rather than complete words in the following paragraphs
- Figure 2, assigning numbers/letters (a, b, …) to the figure is missing
- line 222, 1000 → 1,000
- line 225, MIP2-FAL → MIP2-FAL.
- line 272, 601 ±2 nm → 601 ± 2 nm
Author Response
In this manuscript, the authors proposed a SPR probe based optical sensor for 2-FAL detection in insulating oil samples. On the whole, this article is interesting and well-written. In my opinion, a minor revision needs to be made before acceptance.
- Compared to other sensing methods for the determination of 2-FAL, What is the advantage of SPR sensor?
Reply
SPR sensors belong to the family of optical sensors with intrinsic advantages of high immunity to electromagnetic interferences, high flexibility and feasibility of remote sensing when implemented in optical fibers. Optical techniques for 2-FAL detection are mainly based on measures of the intensity of an absorption peak, but with an intrinsic low signal-to-noise ratio. Alternatively, spectroscopic detection in the UV-visible range was proposed, but one of the main drawbacks is the high interference due to the oil matrix; more recently, the Near Infrared range has been investigated by others, but this method shows a low sensitivity, enabling the detection of 2-FAL only at high concentration (>10 ppm). SPR sensors are based on the resonant phenomenon; therefore, they intrinsically show a high S/N ratio. The high specificity and sensitivity are provided by the application of the specific receptor (the Molecularly Imprinted Polymer, in this paper).
We added more information about this point in the revised text. In particular, some more comments are added to the par. “Discussion and conclusion”, and in other parts of the revised text.
- Figure 5, what is S0, S1, S2, …? Are they some parallel samples?
Reply
The meaning of the symbols has been added in the caption.
- Line 278-285, 241-256, 228-238, what is the reason to create many very short subparagraphs?
Reply
The subparagraphs have been eliminated as suggested.
- What is the background level of 2-FAL in your sample?
Reply
The standards were prepared in fresh (virgin) mineral oil not containing any 2-FAL as determined by HPLC methods. This is the background level in our samples.
- Have you ever evaluated the accuracy of the sensor by recovery experiments?
Reply
We have reported some recovery experiments at different concentration levels on lines 476-482.
- Some formatting issues are expected to be corrected. Just some examples below:
Reply
- line 3, oil. → oil; line 5, Pesavento4,+. → Pesavento4,+; line 13, Correspondence → + Correspondence
- line 26, mg l-1 → mg L-1; line 298, 5 mg l-1 → 5 mg L-1
- line 74, plasmonic resonance (SPR) → SPR; line 141, 2-furaldehyde (2-FAL, CAS N. 98–01–1) → 2-FAL (CAS No. 98-01-1). All the abbreviations should be stated when the corresponding word first appears in the manuscript and always use the abbreviations rather than complete words in the following paragraphs
- Figure 2, assigning numbers/letters (a, b, …) to the figure is missing
- line 222, 1000 → 1,000
- line 225, MIP2-FAL → MIP2-FAL.
- line 272, 601 ±2 nm → 601 ± 2 nm
We modified the text as suggested, in particular, we changed the small l with capital L when required, and in Figures and Tables, and we corrected Fig. 2.
Reviewer 2 Report
Comments and Suggestions for Authors
The manuscript under consideration delves into a novel technique that leverages the phenomenon of Surface Plasmon Resonance (SPR) in an innovative way for bio/chemical applications. In this technique, the receptor part is separated from the detection part, which is a novel innovation. However, several aspects need further clarification and elaboration:
1. Figure 1 only shows the schematic of the detector part, specifically the SPR probe with two variations: without SiOx and with SiOx. The schematic of the POF-MIP (Plastic Optical Fiber-Molecularly Imprinted Polymer) part is not depicted. This omission needs to be addressed to help readers understand the overall system. The structure of the core and cladding of the POF, and how it relates to the MIP, should be illustrated. Additionally, a depiction of the micro-trench MIP chip would be beneficial for contextual understanding.
2. In this research, the change in the refractive index of the medium in contact with the SPR surface causes a shift in resonance conditions, which in turn alters the shape and wavelength of the SPR dip. An illustration showing these modal changes in light should be included to better elucidate this phenomenon for the readers.
3. MIP in massive form? Clarification is needed on whether the MIP is in bulk form or as a thin layer. This distinction is crucial for understanding the interaction dynamics and sensitivity of the SPR system.
4. Sub-section 3.7 (Measurement) requires enhancement with an illustrative measurement diagram. This would help clarify the dielectric conditions (constant refractive index) on the SPR surface and the specific target being measured. Such an illustration would greatly aid in visualizing the experimental setup and the measurement process.
5. In Figure 5, there is a legend labeled s0 to s7. However, these labels are not explained in the narrative. Detailed explanations of these labels should be provided in the manuscript to ensure that readers can accurately interpret the data presented.
Author Response
The manuscript under consideration delves into a novel technique that leverages the phenomenon of Surface Plasmon Resonance (SPR) in an innovative way for bio/chemical applications. In this technique, the receptor part is separated from the detection part, which is a novel innovation. However, several aspects need further clarification and elaboration:
- Figure 1 only shows the schematic of the detector part, specifically the SPR probe with two variations: without SiOx and with SiOx. The schematic of the POF-MIP (Plastic Optical Fiber-Molecularly Imprinted Polymer) part is not depicted. This omission needs to be addressed to help readers understand the overall system. The structure of the core and cladding of the POF, and how it relates to the MIP, should be illustrated. Additionally, a depiction of the micro-trench MIP chip would be beneficial for contextual understanding.
Reply
A schematic view of the POF-MIP chip has been added to Fig 1, for better clarity. The detailed preparation of a similar capturing chip has been reported in previous works, and it is here summarized in paragraph 3.4, which has been modified according to the observation of the reviewer.
- In this research, the change in the refractive index of the medium in contact with the SPR surface causes a shift in resonance conditions, which in turn alters the shape and wavelength of the SPR dip. An illustration showing these modal changes in light should be included to better elucidate this phenomenon for the readers.
Reply
Fig. 1 has been modified in order to better illustrate the concept of the multimodal propagation, and of the different effects of a change of RI of the propagating medium.
- MIP in massive form? Clarification is needed on whether the MIP is in bulk form or as a thin layer. This distinction is crucial for understanding the interaction dynamics and sensitivity of the SPR system.
Reply
With “massive” we meant that the MIP completely fills the microtrench, either as a monolith, or as particles in tight contact with each other. This view is supported by the relatively large amount of prepolymeric mixture which is introduced in the micro-trench and by the picture in Fig. 3. We substituted the term with bulky, opposite to the thin layer concept.
- Sub-section 3.7 (Measurement) requires enhancement with an illustrative measurement diagram. This would help clarify the dielectric conditions (constant refractive index) on the SPR surface and the specific target being measured. Such an illustration would greatly aid in visualizing the experimental setup and the measurement process.
Reply
We illustrated the measurement procedure in a more schematic way (see section 3.7).
- In Figure 5, there is a legend labeled s0 to s7. However, these labels are not explained in the narrative. Detailed explanations of these labels should be provided in the manuscript to ensure that readers can accurately interpret the data presented.
Reply
An explanation has been added to Figure 5.
Round 2
Reviewer 2 Report
Comments and Suggestions for Authors
The authors have addressed the suggested revisions.